# LEANER TRANSFORMERS:
# MORE HEADS, LESS DEPTH

## ABSTRACT

Transformers have reshaped machine learning by leveraging attention to capture complex dependencies, driving major advances across domains. Their success has fueled the belief that ever-larger models are required for strong performance. In this paper, we challenge this assumption by showing that many transformers are unnecessarily oversized. We present a theoretical principle that redefines the role of multi-head attention, demonstrating that multiple heads improve the conditioning of the Jacobian of the attention block. Guided by this insight, we redesign popular architectures with more heads and fewer layers. This trade-off reduces parameter counts by up to 30-50% while preserving accuracy, yielding leaner yet equally effective models. We validate our approach across a range of transformer-based architectures and scales, showing consistent benefits on tasks in computer vision (ImageNet-1k) and language and sequence modeling (GLUE, TinyStories, and the Long-Range Arena benchmark).

## 1 INTRODUCTION

Transformer architectures introduced by Vaswani (2017) have become the dominant architecture across a wide range of fields, including natural language processing (NLP) (Vaswani, 2017; Devlin et al., 2018; Zhuang et al., 2021; Zhen et al., 2022), computer vision (Dosovitskiy et al., 2020; Carion et al., 2020; Liu et al., 2021; Touvron et al., 2021), and robotics (Fu et al., 2024; Maiti et al., 2023; Salzmann et al., 2020). At the heart of their success lies the attention mechanism, which dynamically assigns relevance scores to input elements, enabling the model to capture complex dependencies in data more effectively than traditional architectures.

As transformers continue to scale, the prevailing belief is that heavy overparameterization is necessary for strong performance. Standard designs increase capacity through three main avenues: (1) expanding the number of attention heads, (2) widening feedforward layers, and (3) deepening the network with more layers. While the roles of width and depth have been extensively studied in the context of optimization and generalization (Agarwal et al., 2021; Arora et al., 2018; Zhou & Feng, 2018; Kabkab et al., 2016; Li et al., 2018; Liu et al., 2022; Jacot et al., 2018), the trade-offs involving attention heads remain comparatively underexplored (Levine et al., 2020b;a; Petty et al., 2023; Sanford et al., 2023).

In this paper, we revisit the conventional design of transformers and ask whether current architectures are structured optimally. We introduce a theoretical principle showing that multi-head attention improves the conditioning of the Jacobian of attention layers, lowering its condition number and thereby stabilizing gradient-based optimization (Nocedal & Wright, 1999). Guided by this insight, we demonstrate that attention heads can often be traded for depth, enabling leaner architectures that maintain both optimization stability and accuracy. Since each layer contributes significantly to parameter count, this trade-off offers substantial reductions in model size and memory without compromising performance.

We validate our findings by modifying and retraining a range of existing models in both vision and NLP. Across tasks such as ImageNet-1k classification (Steiner et al., 2021), language modelling on TinyStories (Eldan & Li, 2023), GLUE benchmark evaluation (Wang et al., 2018), and long-context modeling with the LRA benchmark (Tay et al., 2021), we show that transformers redesigned with more heads and fewer layers consistently match or exceed the performance of their original coun-

terparts. These results raise important questions about heavy overparameterization in transformers and highlight opportunities for more principled architecture design.

Our contributions are summarized as follows:

1. A theoretical framework offering a new perspective on multi-head attention, showing that one of its central roles is to improve the conditioning of attention layers.

2. An empirical design principle derived from this theory, demonstrating that depth can be traded for additional heads to reduce parameter count without sacrificing accuracy.

3. A comprehensive empirical validation, confirming the effectiveness of this trade-off on widely used benchmarks.

## 2 RELATED WORK

**Efficient attention-based architectures.** Numerous approaches have been proposed to enhance the efficiency and effectiveness of transformers, particularly by reducing the computational complexity of the attention layer. DeiT (Data-Efficient Image Transformer) (Touvron et al., 2021) improves training efficiency by leveraging distillation tokens, enabling strong performance with significantly fewer data requirements. XCiT (Cross-Covariance Image Transformer) (Ali et al., 2021) introduces a novel attention mechanism that operates on spatial feature cross-covariances, improving feature interactions while substantially reducing computational overhead. VOLO (Vision Outlooker) (Yuan et al., 2022) incorporates outlook attention, which efficiently captures long-range dependencies, outperforming traditional vision transformers (ViTs) while maintaining computational efficiency. Nyströmformer (Xiong et al., 2021b) tackles the quadratic complexity of self-attention using a Nyström-based approximation, reducing it to near-linear time while preserving key attention properties. Other efficient transformer variants have further addressed attention-related bottlenecks. Linformer (Wang et al., 2020) approximates self-attention with low-rank projections, achieving linear complexity by compressing the sequence length dimension. Performer (Choromanski et al., 2021) employs kernelized attention with random feature projections, enabling scalable attention with linear time complexity. Reformer (Kitaev et al., 2020) utilizes locality-sensitive hashing to significantly reduce memory and computational costs, making attention efficient even for long sequences. We take a different approach, exploring whether the inherent complexity of transformers can be reduced to create more compact models that maintain strong performance. Our insights on conditioning are orthogonal to the above methods and demonstrate benefits on several of the aforementioned architectures (ViTs, Nyströmformers).

**Conditioning.** The existing literature provides theoretical support for improved performance of MLPs with better-conditioned weight matrices trained with gradient descent. Liu et al. (2022) used the Neural Tangent Kernel (NTK) framework (Jacot et al., 2018) to show that increasing network width reduces the NTK's condition number, leading to better convergence. As MLPs widen, their weight matrices enter the regime described in theorem 3.1 where the condition number approaches 1. By direct application of the chain rule, this implies that the improved conditioning of the weight matrices leads to a better-conditioned NTK. A vast literature has also explored the roles of width and depth (Poole et al., 2016; Vardi et al., 2022). Arora et al. (2018) showed that, in linear MLPs, depth serves as a preconditioner for stochastic gradient descent, improving optimization as depth increases. Similarly, Agarwal et al. (2021) found that depth enhances the conditioning of non-linear MLPs, provided that activations are properly normalized, thereby facilitating better convergence with gradient-based algorithms. The above studies underscore the importance of both width and depth in achieving good optimization for MLPs. A similar theoretical understanding for transformers is lacking (Levine et al., 2020b;a; Petty et al., 2023; Sanford et al., 2023) and our work helps fill this gap. We reveal a crucial role of multi-head attention in the optimization of transformers and explore its empirical relationship with model depth.

## 3 THEORETICAL FINDINGS

### 3.1 PRELIMINARIES AND NOTATION

**Transformers.** We first briefly review the the transformer architecture (Vaswani, 2017; Dosovitskiy et al., 2020; Prince, 2023). A transformer is composed of stacked layers, also known as "transformer blocks". Each layer is formally represented as a mapping

$$\mathbf{T} : \mathbb{R}^{N \times D} \to \mathbb{R}^{N \times D} \tag{1}$$

defined by the expression

$$\mathbf{T}(X) = \mathbf{F}(\mathbf{A}(X) + X). \tag{2}$$

The dimension $N$ is generally the number of tokens and $D$ is the token embedding dimension (Prince, 2023). The component $\mathbf{F}$ denotes a feedforward multi-layer perceptron (MLP, typically with one hidden layer and a residual connection), and $\mathbf{A}$ represents the self-attention mechanism. In general, depending on the application after stacking such components $\mathbf{T}$ together in a feedforward manner the transformer may have a final head $\mathbf{H}$ that consists of a MLP. For example, for classification $\mathbf{H}$ would be the final classifying head, see Prince (2023) for details.

The self-attention mechanism $\mathbf{A}$ uses three learnable matrices, the query $(Q)$, key $(K)$, and value $(V)$ matrices. Given an input sequence $X \in \mathbb{R}^{N \times D}$, the matrices are first applied as follows: $q = QX$, $k = KX$, $v = VX$, where $Q, K \in \mathbb{R}^{D \times d}$ and $V \in \mathbb{R}^{D \times d}$. These are then combined to produce the output of the self-attention head as follows:

$$\mathbf{A}(X) := \mathbf{softmax} \left( \frac{q\,k^\top}{\sqrt{d}} \right) v \tag{3}$$

where the softmax is applied row-wise, $d$ is known as the head dimension and the scaling $\frac{1}{\sqrt{d}}$ is so that the softmax values don't saturate (Vaswani, 2017). In general, eq. (3) computes the query-value product by a dot-product $qk^\top$. However, more general forms of attention exist that replace the dot-product with a general similarity measure $\phi$ and thus compute an attention block using $\phi(q, k)$: $\mathbf{A}(X) := \mathbf{softmax}(\phi(q, k))\,v$. In this paper, our theoretical claims will be given in the context of self-attention so that we can provide concrete formulas. However, our insights hold true for general forms of attention as we will see in the experiments section 4.

Equation (3) defines the formula for one attention head within a transformer block. In general, transformers use multiple heads leading to multi-head attention. Given an integer $h > 0$, known as the number of heads, we have an attention matrix $\mathbf{A}_i$ for each $1 \le i \le h$, each of dimension $N \times d$. Their outputs are then concatenated together to form a block of attention matrices

$$[\mathbf{A}_1, \cdots, \mathbf{A}_h] \tag{4}$$

where each $\mathbf{A}_i$ is known as an attention head. Note that as each head has dimension $N \times d$ and there are $h$ heads we have that the block $[\mathbf{A}_1, \cdots, \mathbf{A}_h] \in \mathbb{R}^{N \times hd}$. It's often the case that $d = \frac{D}{h}$ and $h$ and thus $h$ must divide $D$. Finally, applying a projection matrix $\mathbf{P} \in \mathbb{R}^{hd \times D}$ we obtain multi-head attention

$$\mathbf{MH} = [\mathbf{A}_1, \cdots, \mathbf{A}_h] \cdot \mathbf{P} \tag{5}$$

giving $\mathbf{MH} \in \mathbb{R}^{N \times D}$.

The attention head of a layer in a transformer $\mathbf{A}(X) \in \mathbb{R}^{N \times d}$ has parameters given by those parameters in $X$ from the previous layer and those given by $Q$, $K$ and $V$ that define $\mathbf{A}(X)$, see eq. (3). Our work will consider the Jacobian of $\mathbf{A}(X)$ with respect to the parameters within the layer of $\mathbf{A}(X)$, namely $Q$, $K$ and $V$. Therefore, when we speak of the Jacobian of $\mathbf{A}(X)$ it will be with respect to $Q, K, V$. We will denote this Jacobian by $\mathbf{J}(\mathbf{A}(X))$ and note that it is defined by

$$\mathbf{J}(\mathbf{A}(X)) = \left[ \frac{\partial \mathbf{A}(X)}{\partial Q}, \frac{\partial \mathbf{A}(X)}{\partial K}, \frac{\partial \mathbf{A}(X)}{\partial V} \right]^\top \tag{6}$$

where each of $\frac{\partial \mathbf{A}(X)}{\partial Q}$, $\frac{\partial \mathbf{A}(X)}{\partial K}$ and $\frac{\partial \mathbf{A}(X)}{\partial V}$ has dimension $(Nd) \cdot (Dd)$. Therefore, $J(\mathbf{A}(X))$ has dimension $(3Dd) \times (Nd)$.

Given a matrix $M \in \mathbb{R}^{m \times n}$ we denote the vectorization of $M$ by $\mathbf{vec}(M) \in \mathbb{R}^{mn \times 1}$ (Magnus & Neudecker, 2019). Note that for such a matrix there is a transformation $T_{mn} \in \mathbb{R}^{mn \times mn}$ such that $T_{mn}\mathbf{vec}(M) = \mathbf{vec}(M^\top)$ where $M^\top$ denotes the transpose of $M$. The matrix $T_{mn}$ is known as a commutation matrix and is a permutation matrix (Magnus & Neudecker, 2019). The maximum singular value of a matrix $M$ will be denoted by $\sigma_{\max}(M)$ and the minimum singular value by $\sigma_{min}(M)$. We will use the standard terminology SVD to denote the singular value decomposition of a matrix. Given a vector $z \in \mathbb{R}^n$ the notation $||z||_2$ will denote the vector 2-norm of $z$. Finally we will need the notion of a $\limsup$ which we remind the reader is the largest value that a sequence gets arbitrarily close to infinitely often, capturing its long-term upper bound behavior (Dym, 2004).

**Condition number.** The condition number of a matrix is the ratio of its largest to smallest singular values. In gradient-based optimization of linear and non-linear systems, the condition number serves as a quantitative measure of how well the optimizer will converge. Lower values indicate a more stable and efficient convergence (Nocedal & Wright, 1999).

**Definition 3.1.** The **condition number** of a full-rank, $n \times m$ matrix $M$ is defined as $\kappa(M) := \sigma_1(M) / \sigma_k(M)$, with the singular values $\sigma_1(M) \geq \cdots \geq \sigma_k(M)$ and $k = \min(m, n)$. Since $M$ is of full rank, all singular values are positive and the condition number is thus well defined. And since $\sigma_1(M) \geq \sigma_k(M)$, the condition number satisfies $\kappa(M) \geq 1$.

## 3.2 MAIN THEORETICAL RESULTS

In this section we provide our main theorem of the paper. From eq. (4), we saw that each attention head has dimension $N \times d$, where $N$ is the numbers of tokens, $D$ is the token embedding dimension. Concatenating $h$ such heads and then applying a projection we produced the final multi-head attention block $\mathbf{MA}$ of dimension $N \times D$. The following lemma shows how the Jacobian of $\mathbf{MA}$ can be determined in terms of the Jacobian of each head $\mathbf{A}_i$ and the projection matrix $\mathbf{P}$.

**Lemma 3.1.** *Let* $\mathbf{MA} \in \mathbb{R}^{N \times D}$ *denote a multi-head attention layer of a transformer block as in eq. (4). Then*

$$\mathbf{J}(\mathbf{MH}) = \begin{bmatrix} \mathrm{Diag}(\mathbf{J}(\mathbf{A_1}), \ldots, \mathbf{J}(\mathbf{A_h}))(\mathbf{P} \otimes I_N) \\ ([\mathbf{A}_1, \ldots, \mathbf{A}_h]^\top \otimes I_D) \end{bmatrix} \in \mathbb{R}^{(3hDd + Dhd) \times ND} \tag{7}$$

The proof of lemma 3.1 is given in section A.1.1. We can now give the statement of the main theorem of this paper.

**Theorem 3.1.** *Let* $\mathbf{MA} \in \mathbb{R}^{N \times D}$ *denote a multi-head attention layer of a transformer block as in eq. (4) built via $h$ attention heads $\mathbf{A}_i \in \mathbb{R}^{N \times d}$ for $1 \leq i \leq h$ and a projection $\mathbf{P} \in \mathbb{R}^{hd \times D}$. Assume, uniformly in $h > 0$, there exists constants $C_1, C_2 > 0$ and $C_3, C_4 > 0$ such that*

$$0 < C_1 \leq \sigma_{\min}(\mathbf{A}_i) \leq \sigma_{\max}(\mathbf{A}_i) \leq C_2 \text{ for } 1 \leq i \leq h \tag{8}$$
$$0 < C_3 \leq \sigma_{\min}(\mathbf{P}) \leq \sigma_{\max}(\mathbf{P}) \leq C_4 \tag{9}$$

*and the columns of $[\mathbf{A}_1 \ldots, \mathbf{A}_h]$ are independent, mean zero, Gaussian distributed entries with covariance $\Sigma$ satisfying $0 < \lambda_{\min}(\Sigma) \leq \lambda_{\max}(\Sigma)$. Then with probability approaching 1 we have the following asymptotic bound*

$$\limsup_{h \to \infty} \kappa(\mathbf{J}(\mathbf{MA}) \leq \sqrt{\frac{\lambda_{\max}(\Sigma)}{\lambda_{\min}(\Sigma)}}. \tag{10}$$

*Furthermore, if we assume that the columns of $[\mathbf{A}_1, \ldots, \mathbf{A}_h]$ are isotropic, so that $\Sigma = \sigma^2 I_N$, then with probability approaching 1 we have*

$$\kappa(\mathbf{J}(\mathbf{MA})) = 1 + \mathcal{O}\left(\sqrt{\frac{N}{hd}}\right) \text{ as } h \to \infty. \tag{11}$$

*In particular, for large $h$ we have that $\kappa(\mathbf{J}(\mathbf{MA})) \approx 1$.*

The theorem highlights that, while the Jacobian of an individual attention head $\mathbf{J}(\mathbf{A}_i)$ of dimension $Nd \times Nd$ may not be well-conditioned, the *concatenation* of multiple heads leads to a well-conditioned Jacobian of a multi-head layer, under some assumptions. The proof of theorem 3.1 is given in section A.1.1. For a discussion of the assumptions in theorem 3.1 we refer the reader to section A.1.1.

**Why scale the number of heads?**  Beyond spectral considerations, increasing the number of heads provides a useful inductive bias: more heads yield *more distinct attention patterns*. With a moderate per-head dimension $d$ (e.g., 64-128), each head can specialize to different spatial correlations in the token sequence, effectively offering multiple "views" of the data. By contrast, enlarging $d$ within a single head makes that head richer but does not multiply the number of patterns. In practice, this multi-view bias from additional heads often proves more effective across tasks.

> **Implementation.** In many transformers the dimensions $D$, $d$, and $h$ (see section 3.1) satisfy $d = D/h$. Thus, increasing $h$ can make the per-head width $d$ very small. In our experiments (section 4), we instead fix the model width $D$ and set the per-head size to $d \in \{64, 128\}$; we then increase $h$ without enforcing $d = D/h$.

## 3.3 TRADING DEPTH FOR HEADS

The literature reviewed in section 2 indicates that depth and width play complementary roles in the optimization of neural networks. Building on this, Theorem 3.1 shows that increasing the number of heads improves the conditioning of the multi-head Jacobian, which should facilitate optimization. At the same time, adding heads increases model capacity, which can itself improve performance. Our interest is in whether these benefits can be harnessed to *trade* depth for heads: can we use more heads to maintain accuracy while reducing the number of layers?

> **Hypothesis (trading depth for heads).** For a fixed token embedding dimension $D$ and per–head dimension $d$, increasing the number of attention heads $h$ improves optimization, via more diverse and better-conditioned attention mappings, so that the number of transformer layers $L$ can be reduced without degrading performance.

> **Motivation.** A single transformer layer typically carries a large parameter budget, dominated by the feed-forward block, whereas adding several heads within a layer increases parameters more modestly. Thus, reducing depth can quickly shrink model size, and additional heads may preserve performance by providing richer, complementary attention patterns. In section 4 we evaluate this hypothesis across architectures and tasks. A full theoretical account of when (and why) depth can be traded for head multiplicity remains open. Our results suggest concrete directions for future work on optimal architecture design.

## 4 EXPERIMENTS

We perform extensive experiments with a variety of transformer-based models. Our goals are (1) to empirically verify the prediction of theorem 3.1 about improvements in conditioning and (2) to evaluate the downstream benefits on a variety of different transformer architectures and applications.

## 4.1 IMAGE CLASSIFICATION

We consider standard vision transformers (ViTs) from the literature. We modify their architecture according to the findings from section 3 and re-train them from scratch on ImageNet-1k (Steiner et al., 2021). Our approach enables reductions in parameter count by up to 30-50% of existing models without compromising their accuracy. The explicit training details, implementation and hardware used for all experiments in this subsection can be found in section A.2.1.

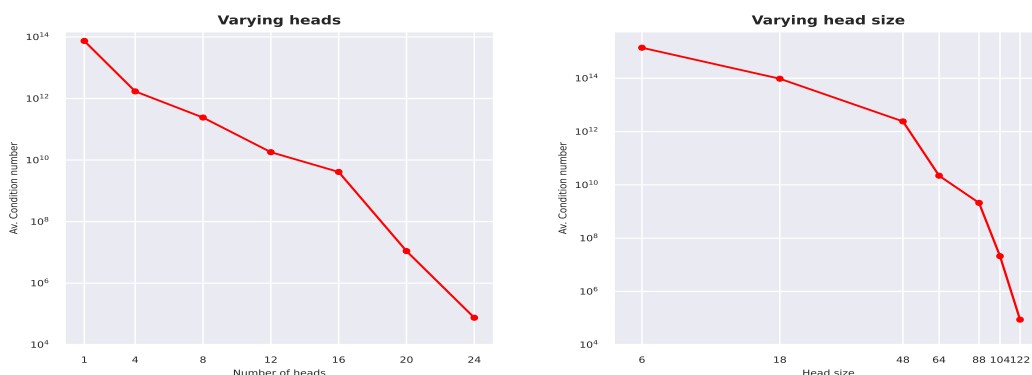

Figure 1: Empirical measurement of the condition number of the attention layers in ViT-Bs with different numbers of heads (left) and varying head dimension (left). In both cases the condition number of the Jacobian improves (lower number) following the predictions of theorem 3.1 and theorem A.2.

### 4.1.1 STANDARD ViTs

We adopt the ViT-Base (ViT-B) architecture (Dosovitskiy et al., 2020), a widely used model for image classification. An input image is divided into non-overlapping $16!\times!16$ patches, which are linearly projected into 768-dimensional token embeddings that serve as input to the transformer layers. ViT-B consists of 12 layers, each with 12 attention heads of dimension 64 (so that $12 \times 64 = 768$, matching the embedding size). The MLP blocks use hidden layers of size $4 \times 768 = 3{,}072$.

**Validating the effects on conditioning.** To validate theorem 3.1, we varied the number of heads in a ViT-B while fixing the head dimension at $d = 64$ and the number of layers at 12, training models on ImageNet-1k with $h \in 1, 4, 8, 12, 16, 20, 24$. Since theorem 3.1 also applies when fixing $h$ and increasing $d$ (see theorem A.2), we performed a second experiment with $h = 12$ and $d \in 6, 18, 48, 64, 88, 104, 122$ to match parameter counts. Each model was trained for 300 epochs, and every 50 epochs we computed the Jacobian condition number of the attention matrix (averaged across layers). As shown in fig. 1, the condition number decreases markedly with more heads, confirming theorem 3.1.

**Varying number of heads.** We first fix the depth at 12 layers and vary the number of heads from 2 to 18 with head dimension 64. To further test the hypothesis from section 3.3, we repeat the experiment with depth reduced to 8 layers. All models are trained on ImageNet-1k for 300 epochs with AdamW, following standard protocols (Steiner et al., 2021; Dosovitskiy et al., 2020) (see section A.2.1). As shown in fig. 2, accuracy consistently increases with more heads: in the 12-layer setting, performance surpasses the original baseline once $h > 12$, while in the 8-layer setting, accuracy remains strongly correlated with head count. Importantly, reducing depth offsets the parameter cost, with models above 12 heads outperforming the baseline using substantially fewer parameters (61.2–67.4M vs. 86.6M).

**Varying head dimension.** Next, we fix the number of heads at $h = 12$ and vary the head dimension over $d \in \{10, 18, 32, 48, 54, 64, 80, 88, 96\}$. This setup yields parameter counts directly comparable to those in the head-variation experiments. As shown in fig. 3 (left), accuracy improves consistently as the head dimension increases. Repeating the experiment with a reduced depth of 8, we observe that for $d \in \{88, 96\}$ the model slightly outperforms the standard ViT-B while using fewer parameters.

**Varying heads vs. head dimension.** Table 1 reports training times when varying head dimension $d$ with fixed heads $h = 12$, compared with varying heads $h$ while fixing $d = 64$, for a depth of 12. The table highlights a clear advantage for varying heads: models train faster for the same parameter budget compared to varying head dimension. Furthermore, comparing fig. 3 with fig. 2, we find that increasing heads at depth 8 produces three configurations that outperform all head-

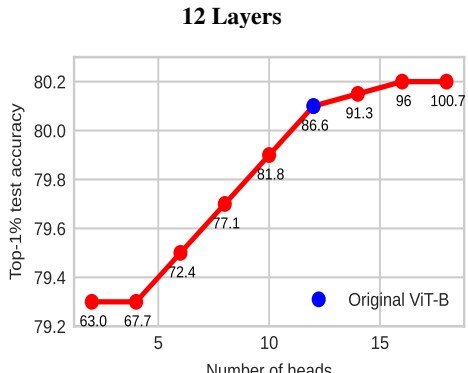
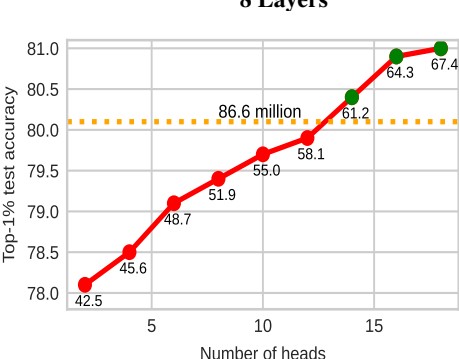

Figure 2: Empirical measurement of the condition number of the attention layers in ViT-Bs with different numbers of heads (left) and varying head dimension (left). In both cases the condition number of the Jacobian improves (lower number) following the predictions of theorem 3.1 and theorem A.2.

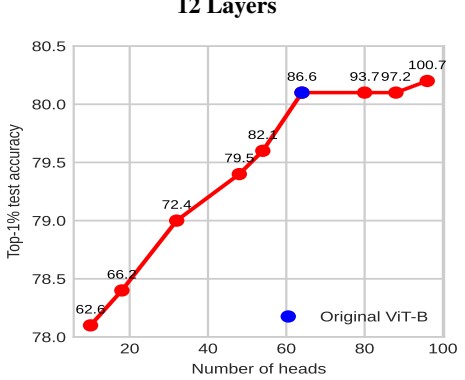
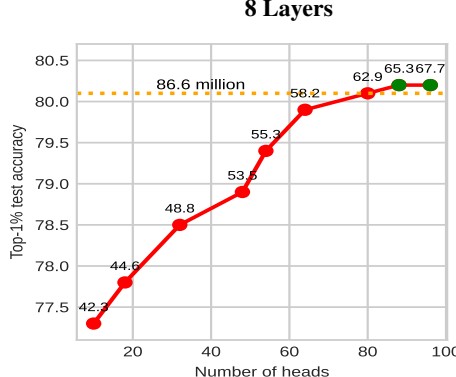

Figure 3: Empirical measurement of the condition number of the attention layers in ViT-Bs with different numbers of heads (left) and varying head dimension (left). In both cases the condition number of the Jacobian improves (lower number) following the predictions of theorem 3.1 and theorem A.2.

dimension variants. Overall, in parameter-matched comparisons, varying the number of heads yields both shorter training times and stronger accuracy. Therefore, **in this paper we focus primarily on varying heads**.

Table 1: Rows 2-4 report the effect of varying the head dimension $d$ while keeping the number of heads $h$ fixed. Rows 5-7 instead vary $h$ while holding $d$ fixed. Row is the standard ViT-B baseline. All models use 12 layers. We observe that adjusting $h$ yields shorter training times for the same parameter budget compared to adjusting $d$.

| Depth | $h$ | $d$ | Acc. (%) | Params. (M) | Time (h:min) |
|-------|-----|-----|----------|-------------|--------------|
| 12 | 12 | 64 | 80.1 | 86 | 29:34 |
| 12 | 12 | 80 | 80.1 | 94 | 35:08 |
| 12 | 12 | 88 | 80.1 | 97 | 39:18 |
| 12 | 12 | 96 | 80.2 | 101 | 45:41 |
| 12 | 14 | 64 | 80.2 | 91 | 31:16 |
| 12 | 16 | 64 | 80.2 | 96 | 33:42 |
| 12 | 18 | 64 | 80.2 | 101 | 36:51 |

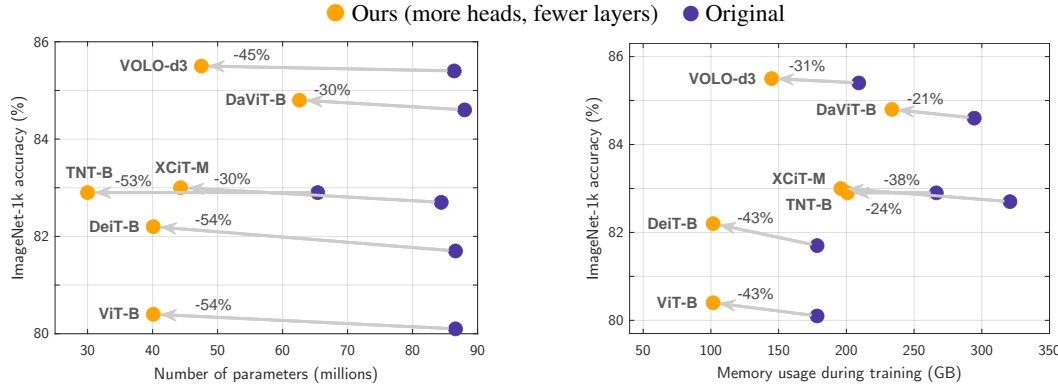

Figure 4: Other vision transformer architectures. Improvements in accuracy relative to parameter count (left) and training memory usage (right). All models benefit significantly from our approach.

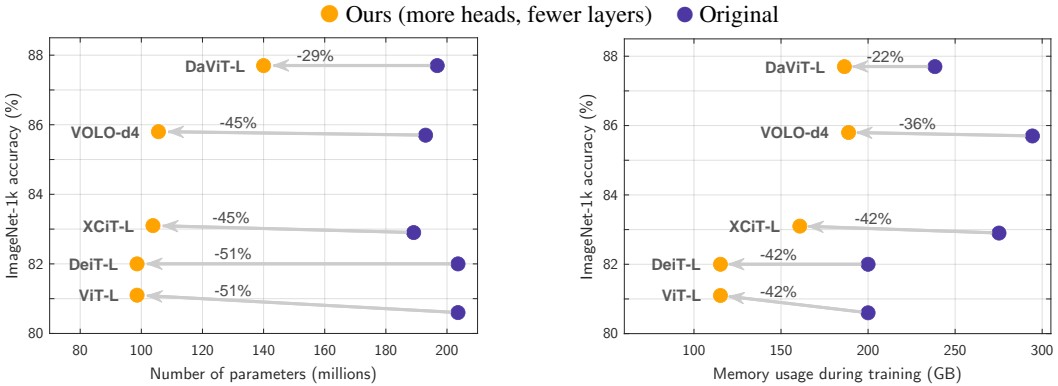

Figure 5: Large vision transformer architectures. We observe consistent improvements in accuracy relative to parameter count (left) and in training memory usage (right), mirroring the trends observed for smaller models.

### 4.1.2 OTHER VISION TRANSFORMERS

We apply our strategy to a variety of alternative transformer-based architectures in the 60-90 M parameter range: DeiT (Touvron et al., 2021), XCiT (Ali et al., 2021), TNT (Han et al., 2021), VOLO (Yuan et al., 2022), and DaViT (Ding et al., 2022), all pretrained on ImageNet-1k. We report our best configurations in fig. 4 . In all cases, reducing depth and increasing the number heads leads to models with similar or higher accuracy with substantial reductions in parameter count. This indicates that many models are unnecessarily oversized. This also corresponds to substantial reductions in memory during training (reported separately in fig. 4). For an ablation, see section A.2. We also evaluate models in the 180 - 200 M parameter range. Figure 5 shows similar improvements in accuracy, parameter count, and memory usage.

**What about width?** In section A.2.1, we analyze variations in the hidden-layer size of the MLP blocks in a ViT-B model as an alternative way to adjust model width, but find that this has only limited impact on performance compared to increasing the number of attention heads.

### 4.2 LANGUAGE MODELING

**Crammed BERT.** We first consider the Crammed-BERT architecture (Geiping & Goldstein, 2023). trained on the Pile dataset (Gao et al., 2021) following Geiping & Goldstein (2023). We evaluate these models on the GLUE benchmark (Wang et al., 2018). The original model uses 12 heads and 16 layers. As hypothesized, we find that increasing the number of heads leads to better performance, so much so that the depth can be reduced and still match the performance of the

Table 2: Comparison of a pretrained original Crammed BERT (16 layers, 12 heads per layer) with our leaner variant (10 layers, 24 heads) on the GLUE benchmark. For each task our learner variant achieves comparable performance with much less parameters.

| | MNLI | SST-2 | STSB | RTE | QNLI | QQP | MRPC | CoLA | GLUE | Params. | Mem. |
|---|---|---|---|---|---|---|---|---|---|---|---|
| Original | **83.8** | 92.3 | 86.3 | 55.1 | 90.1 | 87.3 | 85.0 | 48.9 | 78.6 | 119 M | 13.8 GB |
| Ours | 83.7 | 92.3 | 86.3 | **55.3** | 90.1 | 87.3 | **85.2** | 48.9 | 78.6 | 84 M $(-29\%)$ | 10.3 GB $(-25\%)$ |

Table 3: GPT-2 models trained on the TinyStories dataset. We compare a baseline model with 12 layers and 12 attention heads (Eldan & Li, 2023) and our variant with 4 layers and 16 heads. We achieve superior performance at a much smaller size and memory usage.

| | Val. loss | Parameters | Memory |
|---|---|---|---|
| GPT-2 (original) | 2.47 | 89 M | 12.8 GB |
| GPT-2 (ours) | **2.41** | 64 M (-28%) | 9.7 GB (-24%) |

original model (see table 2). In particular, we find that 24 attention heads and 10 layers produce a compact architecture that performs similarly on GLUE as the original model.

**GPT-2.** We proceed similarly with a GPT-2 architecture trained on the TinyStories dataset (Eldan & Li, 2023). As the original configuration, we use the 12-layer, 12-head model (89 M parameters) from Eldan & Li (2023). We then increase the number of heads to 16 while reducing the depth to 4 layers. As shown in table 3, our variant outperforms the original in both validation loss. Moreover, it achieves these improvements with fewer parameters and reduced memory usage during training.

### 4.3 LONG RANGE SEQUENCE MODELING ON LRA BENCHMARK

Long-range sequence modeling is critical for transformers, allowing them to capture dependencies across hundred/thousands of tokens. To assess our approach in this regime, we use the Long Range Arena (LRA) benchmark, a standard suite for testing long-context modeling (see section A.2.2).

## 5 CONCLUSIONS

In this work, we reexamined the role of multi-head attention in transformers. Our analysis revealed that increasing the number of heads improves the conditioning of the Jacobian of the attention matrices, a finding we confirmed empirically on vision transformers. Building on previous studies of MLP conditioning, we hypothesized that an increase of the number of heads could reduce the depth required to achieve high performance. We tested this on tasks including image classification, language generation, and long sequence modeling, and found that leaner, shallower architectures with more attention heads perform comparably to their deeper counterparts. These results suggest a promising avenue for designing efficient transformers without sacrificing performance.

## 6 LIMITATIONS AND OPEN QUESTIONS

While our results demonstrate that depth can be traded for additional attention heads without loss in performance, a complete theoretical explanation of this balance remains open, and developing quantitative tools to predict such trade-offs is an important direction. Our theorem shows that more heads improve the Jacobian's condition number, but the precise effect on training dynamics and accuracy is supported only empirically, leaving a deeper understanding of this link as an open challenge. Finally, our experiments were limited to models of up to $\sim$200M parameters due to resource constraints, it remains to be seen whether the observed benefits persist at larger scales such as $\sim$1B. [1]

---

[1]Digital writing assistance tools were used for grammar and formatting. No large language models were involved in the research itself, and all scientific contributions are original work by the authors.

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

# A  APPENDIX

## ETHICS STATEMENT

All experiments in this study were conducted on publicly available benchmark datasets. No human subjects, personal information, or sensitive data were used. The methods introduced are intended purely for fundamental research in machine learning.

## REPRODUCIBILITY STATEMENT

We have taken care to ensure the reproducibility of all results presented in this paper. Where external code was used, explicit references are provided, and all experimental settings, including hardware details, are documented in the appendix. In addition, full proofs of all theoretical results are included in the appendix to enable independent verification.

## USE OF LLMS

We used digital writing assistance tools for grammar and formatting. No large language models were involved in conducting the research, and all scientific contributions are the original work of the authors.

## A.1  THEORETICAL RESULTS

### A.1.1  PROOF OF RESULTS FROM SECTION 3

In this section we give the proofs of lemma 3.1 and theorem 3.1. In order to prove theorem 3.1, we will need an auxiliary theorem that provides bounds on the singular values of the Jacobian of a multi-head attention layer which is given by theorem A.1.

We will start with the proof of lemma 3.1.

*Proof of lemma 3.1.* We break the proof up into a number of steps. We use two standard vectorized Kronecker identities. For arbitrary matrices $A, B, C$:

$$\text{vec}(AB) = (I \otimes A)\,\text{vec}(B), \qquad \text{vec}(AB) = (B^\top \otimes I)\,\text{vec}(A). \tag{12}$$

**Step 1: Vectorize the multi-head output.**  Let $\mathbf{A}_{\text{cat}} = [\mathbf{A_1}, \ldots, \mathbf{A}_h]$. By eq. (12),

$$y = \text{vec}(\mathbf{A}_{\text{cat}}\mathbf{P}) = (\mathbf{P}^\top \otimes I_N)\,\text{vec}(\mathbf{A}_{\text{cat}}) = (I_D \otimes \mathbf{A}_{\text{cat}})\,\text{vec}(\mathbf{P}). \tag{13}$$

The first equality will be used for derivatives w.r.t. the *head* parameters (which affect $\mathbf{A}_{\text{cat}}$); the second for derivatives w.r.t. $\mathbf{P}$.

**Step 2: Jacobian w.r.t. P.**  From the second representation in eq. (13), $y = (I_D \otimes \mathbf{A}_{\text{cat}})\,\text{vec}(\mathbf{P})$. In row–Jacobian form this gives $\frac{\partial y}{\partial\,\text{vec}(\mathbf{P})} = I_D \otimes \mathbf{A}_{\text{cat}} \in \mathbb{R}^{(ND)\times(Dhd)}$. Transposing yields the column–Jacobian block

$$\frac{\partial\,\text{vec}(\mathbf{P})}{\partial\,y} = \left(I_D \otimes \mathbf{A}_{\text{cat}}\right)^\top \in \mathbb{R}^{(Dhd)\times(ND)}.$$

**Step 3: Jacobian w.r.t. head parameters.**  From the first representation in eq. (13), $y = (\mathbf{P}^\top \otimes I_N)\,\text{vec}(\mathbf{A}_{\text{cat}})$. Differentiate w.r.t. the stacked head parameters $[\theta_1; \ldots; \theta_h]$. Only the $i$-th block $\mathbf{A}_i$ depends on $\theta_i$, hence

$$\frac{\partial\,y}{\partial\,[\theta_1; \ldots; \theta_h]} = \left(\mathbf{P}^\top \otimes I_N\right)\text{Diag}\!\left(\mathbf{J}(\mathbf{A}_1)^\top, \ldots, \mathbf{J}(\mathbf{A}_h)^\top\right) \in \mathbb{R}^{(ND)\times(3hDd)}.$$

Transposing gives the desired column–Jacobian block:

$$\frac{\partial\,[\theta_1; \ldots; \theta_h]}{\partial\,y} = \text{Diag}\!\left(\mathbf{J}(\mathbf{A}_1), \ldots, \mathbf{J}(\mathbf{A}_h)\right)\left(\mathbf{P} \otimes I_N\right) \in \mathbb{R}^{(3hDd)\times(ND)}.$$

**Step 4: Stack the blocks.** By definition, $\theta = \big[\theta_1; \ldots; \theta_h; \operatorname{vec}(\mathbf{P})\big]$, so stacking the two column–Jacobian blocks from Steps 2–3 yields

$$\frac{\partial \theta}{\partial y} = \begin{bmatrix} \operatorname{Diag}\big(\mathbf{J}(\mathbf{A}_1), \ldots, \mathbf{J}(\mathbf{A}_h)\big) \left(P \otimes I_N\right) \\ \left(I_D \otimes \mathbf{A}_{\mathrm{cat}}\right)^{\top} \end{bmatrix}.$$

**Step 5: Dimensions.** Since $\mathbf{J}(\mathbf{A}_i) \in \mathbb{R}^{(3Dd)\times(Nd)}$, we have $\operatorname{Diag}(\mathbf{J}(\mathbf{A}_1), \ldots, \mathbf{J}(\mathbf{A}_h)) \in \mathbb{R}^{(3hDd)\times(Nhd)}$ and $\mathbf{P} \otimes I_N \in \mathbb{R}^{(Nhd)\times(ND)}$, hence the upper block is $(3hDd)\times(ND)$. Also $I_D \otimes \mathbf{A}_{\mathrm{cat}} \in \mathbb{R}^{(ND)\times(Dhd)}$, so its transpose is $(Dhd)\times(ND)$. Stacking gives a grand column–Jacobian of size $(3hDd + Dhd)\times(ND)$, as claimed. $\qquad\square$

To establish the proof of theorem 3.1, we first require an auxiliary result of our own. This theorem provides explicit bounds on the singular values of the Jacobian associated with the multi-head attention block, and will serve as a key technical tool in the argument that follows.

---

**Theorem A.1.** *Let* $\mathbf{MA} \in \mathbb{R}^{N \times D}$ *denote a multi-head attention layer of a transformer block as in eq.* (4) *built via* $h$ *attention heads* $\mathbf{A}_i \in \mathbb{R}^{N \times d}$ *for* $1 \le i \le h$ *and a projection* $\mathbf{P} \in \mathbb{R}^{hd \times D}$. *Then we have that*

$$\sigma_{\min}(\mathbf{J}(\mathbf{MA})) \ge \sqrt{\sigma_{\min}(\mathbf{P})^2 \bigg( \Big( \min_{1 \le i \le h} \sigma_{\min}(\mathbf{J}(\mathbf{A}_i)) \Big)^2 + (\sigma_{\min}([\mathbf{A}_1, \ldots, \mathbf{A}_h])^2) \bigg)} \qquad (14)$$

$$\sigma_{\max}(\mathbf{J}(\mathbf{MA})) \le \sqrt{\sigma_{\max}(\mathbf{P})^2 \bigg( \Big( \max_{1 \le i \le h} \sigma_{\max}(\mathbf{J}(\mathbf{A}_i)) \Big)^2 + (\sigma_{\max}([\mathbf{A}_1, \ldots, \mathbf{A}_h])^2) \bigg)}. \qquad (15)$$

*Therefore, we have the following condition number bound*

$$\kappa(\mathbf{J}(\mathbf{MA})) \le \sqrt{\frac{\sigma_{\min}(\mathbf{P})^2 \bigg( \Big( \min_{1 \le i \le h} \sigma_{\min}(\mathbf{J}(\mathbf{A}_i)) \Big)^2 + (\sigma_{\min}([\mathbf{A}_1, \ldots, \mathbf{A}_h])^2) \bigg)}{\sigma_{\max}(\mathbf{P})^2 \bigg( \Big( \max_{1 \le i \le h} \sigma_{\max}(\mathbf{J}(\mathbf{A}_i)) \Big)^2 + (\sigma_{\max}([\mathbf{A}_1, \ldots, \mathbf{A}_h])^2) \bigg)}} \qquad (16)$$

---

*Proof.* We break the proof up into a number of steps. Let

$$\mathbf{A}_{\mathrm{cat}} = [\mathbf{A}_1, \ldots, \mathbf{A}_h] \in \mathbb{R}^{N \times (hd)}, \qquad \mathbf{P} \in \mathbb{R}^{(hd)\times D}, \qquad \mathbf{MA} = \mathbf{A}_{\mathrm{cat}}\mathbf{P} \in \mathbb{R}^{N \times D},$$

and for each head $i$,

$$\theta_i = \big[\operatorname{vec}(W_{Q_i}); \operatorname{vec}(W_{K_i}); \operatorname{vec}(W_{V_i})\big] \in \mathbb{R}^{3Dd}, \qquad \mathbf{J}(\mathbf{A}_i) \in \mathbb{R}^{(3Dd)\times(Nd)}.$$

Let $y := \operatorname{vec}(\mathbf{MA}) \in \mathbb{R}^{ND}$ and $\theta := [\theta_1; \ldots; \theta_h; \operatorname{vec}(\mathbf{P})] \in \mathbb{R}^{3hDd + Dhd}$.

**Step 1: A block formula for the Jacobian.** We use the standard vectorization identities, valid for arbitrary matrices $A, B, C$:

$$\operatorname{vec}(AB) = (I \otimes A)\operatorname{vec}(B), \qquad \operatorname{vec}(AB) = (B^{\top} \otimes I)\operatorname{vec}(A). \qquad (17)$$

Applying eq. (17) twice gives two equivalent expressions for $y$:

$$y = \operatorname{vec}(\mathbf{A}_{\mathrm{cat}}\mathbf{P}) = (\mathbf{P}^{\top} \otimes I_N)\operatorname{vec}(\mathbf{A}_{\mathrm{cat}}) = (I_D \otimes \mathbf{A}_{\mathrm{cat}})\operatorname{vec}(\mathbf{P}). \qquad (18)$$

Only the $i$-th block of $\operatorname{vec}(\mathbf{A}_{\mathrm{cat}}) = [\operatorname{vec}(\mathbf{A}_1); \ldots; \operatorname{vec}(\mathbf{A}_h)]$ depends on $\theta_i$; differentiating eq. (18) yields the Jacobian block

$$\operatorname{Diag}\big(\mathbf{J}(\mathbf{A}_1), \ldots, \mathbf{J}(\mathbf{A}_h)\big)(\mathbf{P} \otimes I_N)$$

Likewise, differentiating $y = (I_D \otimes \mathbf{A}_{\mathrm{cat}})\operatorname{vec}(\mathbf{P})$ w.r.t. $\operatorname{vec}(\mathbf{P})$ gives the second Jacobian block $\mathbf{A}_{\mathrm{cat}}^{\top} \otimes I_D$. Altogether,

$$\mathbf{J}(\mathbf{MA}) = \begin{bmatrix} \operatorname{Diag}\big(\mathbf{J}(\mathbf{A}_1), \ldots, \mathbf{J}(\mathbf{A}_h)\big)(\mathbf{P} \otimes I_N) \\ \left(I_D \otimes \mathbf{A}_{\mathrm{cat}}\right)^{\top} \end{bmatrix} \in \mathbb{R}^{(3hDd + Dhd)\times(ND)}. \qquad (19)$$

**Step 2: Singular-value bounds for a vertical stack.** For ease of exposition, write $\mathbf{J} := \mathbf{J}(\mathbf{M}\mathbf{A}) = \begin{bmatrix} U \\ V \end{bmatrix}$ with

$$U := \mathrm{Diag}\big(\mathbf{J}(\mathbf{A}_1), \dots, \mathbf{J}(\mathbf{A}_h)\big) (\mathbf{P} \otimes I_N), \qquad V := \big(I_D \otimes \mathbf{A}_{\mathrm{cat}}\big)^\top.$$

Then $\mathbf{J}\mathbf{J}^\top = UU^\top + VV^\top$. For any unit vector $x$,

$$x^\top (UU^\top + VV^\top)x = x^\top UU^\top x + x^\top VV^\top x \geq \lambda_{\min}(UU^\top) + \lambda_{\min}(VV^\top),$$

hence

$$\sigma_{\min}(\mathbf{J})^2 = \lambda_{\min}(\mathbf{J}\mathbf{J}^\top) \geq \sigma_{\min}(U)^2 + \sigma_{\min}(V)^2. \tag{20}$$

By the operator-norm triangle inequality,

$$\sigma_{\max}(\mathbf{J})^2 = \|UU^\top + VV^\top\|_2 \leq \|UU^\top\|_2 + \|VV^\top\|_2 = \sigma_{\max}(U)^2 + \sigma_{\max}(V)^2. \tag{21}$$

**Step 3: Bounding the factors $U$ and $V$.** We invoke standard singular-value facts: for any conformable $A, B$,

$$\sigma_{\min}(AB) \geq \sigma_{\min}(A)\,\sigma_{\min}(B), \qquad \sigma_{\max}(AB) \leq \sigma_{\max}(A)\,\sigma_{\max}(B). \tag{22}$$

For a block diagonal matrix,

$$\sigma_{\min}\big(\mathrm{Diag}(M_1, \dots, M_h)\big) = \min_i \sigma_{\min}(M_i), \quad \sigma_{\max}\big(\mathrm{Diag}(M_1, \dots, M_h)\big) = \max_i \sigma_{\max}(M_i). \tag{23}$$

For Kronecker products (with $I$ denoting an identity),

$$\sigma_{\min}(A \otimes B) = \sigma_{\min}(A)\,\sigma_{\min}(B), \qquad \sigma_{\max}(A \otimes B) = \sigma_{\max}(A)\,\sigma_{\max}(B). \tag{24}$$

Applying eq. (22)–eq. (24) to $U$ gives

$$\sigma_{\min}(U) \geq \sigma_{\min}\big(\mathrm{Diag}(\mathbf{J}(\mathbf{A}_i)^\top)\big)\,\sigma_{\min}(\mathbf{P} \otimes I_N) = \big(\min_i \sigma_{\min}(\mathbf{J}(\mathbf{A}_i))\big)\,\sigma_{\min}(\mathbf{P}),$$

$$\sigma_{\max}(U) \leq \sigma_{\max}\big(\mathrm{Diag}(\mathbf{J}(\mathbf{A}_i)^\top)\big)\,\sigma_{\max}(\mathbf{P} \otimes I_N) = \big(\max_i \sigma_{\max}(\mathbf{J}(\mathbf{A}_i))\big)\,\sigma_{\max}(\mathbf{P}).$$

For $V = (I_D \otimes \mathbf{A}_{\mathrm{cat}})^\top$, singular values are invariant under transpose and

$$\sigma_{\min}(V) = \sigma_{\min}(I_D \otimes \mathbf{A}_{\mathrm{cat}}) = \sigma_{\min}(\mathbf{A}_{\mathrm{cat}}), \qquad \sigma_{\max}(V) = \sigma_{\max}(I_D \otimes \mathbf{A}_{\mathrm{cat}}) = \sigma_{\max}(\mathbf{A}_{\mathrm{cat}}).$$

**Step 4: Combine.** Substituting these bounds into eq. (20)–eq. (21) yields

$$\sigma_{\min}(J) \geq \sqrt{\sigma_{\min}(\mathbf{P})^2 \Big(\min_i \sigma_{\min}(\mathbf{J}(\mathbf{A}_i))\Big)^2 + \sigma_{\min}(\mathbf{A}_{\mathrm{cat}})^2},$$

$$\sigma_{\max}(J) \leq \sqrt{\sigma_{\max}(\mathbf{P})^2 \Big(\max_i \sigma_{\max}(\mathbf{J}(\mathbf{A}_i))\Big)^2 + \sigma_{\max}(\mathbf{A}_{\mathrm{cat}})^2}.$$

Dividing the second inequality by the first proves the stated condition-number bound for $\mathbf{J}(\mathbf{M}\mathbf{A})$ and completes the proof. $\square$

Using the above theorem A.1 we can give the proof of theorem 3.1.

*Proof of the theorem 3.1.* We will break the proof up into a number of steps. We start by recalling some notation we have been using.

**Setup and notation.** Fix the per–head width $d$ and let $h \in \mathbb{N}$, so that

$$\mathbf{A}_{\mathrm{cat}} = [\mathbf{A}_1, \dots, \mathbf{A}_h] \in \mathbb{R}^{N \times (hd)}, \qquad \mathbf{P} \in \mathbb{R}^{(hd) \times D}, \qquad \mathbf{M}\mathbf{A} = \mathbf{A}_{\mathrm{cat}}\mathbf{P} \in \mathbb{R}^{N \times D}.$$

For each head $i$, collect parameters $\theta_i \in \mathbb{R}^{3Dd}$ and define the (row) Jacobian $\mathbf{J}(\mathbf{A}_i) := \frac{\partial \mathrm{vec}(\mathbf{A}_i)}{\partial \theta_i} \in \mathbb{R}^{(Nd) \times (3Dd)}$. Let $y := \mathrm{vec}(\mathbf{M}\mathbf{A}) \in \mathbb{R}^{ND}$ and $\theta := [\theta_1; \dots; \theta_h; \mathrm{vec}(\mathbf{P})] \in \mathbb{R}^{3hDd + Dhd}$. Recall $\mathbf{J}(\mathbf{M}\mathbf{A})$ is the Jacobian we are analyzing.

**Step 1: Invoking theorem A.1**   By the theorem 3.1 we have the following bounds

$$\sigma_{\min}(\mathbf{J}) \;\geq\; \sqrt{\sigma_{\min}(\mathbf{P})^2\Big(\min_i \sigma_{\min}(\mathbf{J}(\mathbf{A}_i))\Big)^2 \;+\; \sigma_{\min}(\mathbf{A}_{\mathrm{cat}})^2} \tag{25}$$

$$\sigma_{\max}(\mathbf{J}) \;\leq\; \sqrt{\sigma_{\max}(\mathbf{P})^2\Big(\max_i \sigma_{\max}(\mathbf{J}(\mathbf{A}_i))\Big)^2 \;+\; \sigma_{\max}(\mathbf{A}_{\mathrm{cat}})^2}\;. \tag{26}$$

**Step 2: Uniform bounds on per–head Jacobians and the projection.**   By the assumptions eq. (8) and eq. (9), there exist $0 < C_1 \leq C_2 < \infty$ and $0 < C_3 \leq C_4 < \infty$, independent of $h$, such that for all $i$,

$$C_1 \;\leq\; \sigma_{\min}(\mathbf{J}(\mathbf{A}_i)) \;\leq\; \sigma_{\max}(\mathbf{J}(\mathbf{A}_i)) \;\leq\; C_2, \qquad C_3 \;\leq\; \sigma_{\min}(\mathbf{P}) \;\leq\; \sigma_{\max}(\mathbf{P}) \;\leq\; C_4.$$

Substituting these into eq. (25) yields

$$\sigma_{\min}(\mathbf{J}) \;\geq\; \sqrt{C_3^2 C_1^2 + \sigma_{\min}(\mathbf{A}_{\mathrm{cat}})^2}\,, \qquad \sigma_{\max}(\mathbf{J}) \;\leq\; \sqrt{C_4^2 C_2^2 + \sigma_{\max}(\mathbf{A}_{\mathrm{cat}})^2}\;. \tag{27}$$

Hence

$$\kappa(\mathbf{J}) \;=\; \frac{\sigma_{\max}(\mathbf{J})}{\sigma_{\min}(\mathbf{J})} \;\leq\; \sqrt{\frac{C_4^2 C_2^2 + \sigma_{\max}(\mathbf{A}_{\mathrm{cat}})^2}{C_3^2 C_1^2 + \sigma_{\min}(\mathbf{A}_{\mathrm{cat}})^2}}\;. \tag{28}$$

**Step 3: Random–matrix control of $\sigma_{\min/\max}(\mathbf{A}_{\mathrm{cat}})$.**   Recall that we assumed: the $m := hd$ columns of $\mathbf{A}_{\mathrm{cat}}$ are independent, mean–zero, Gaussian in $\mathbb{R}^N$ with common covariance $\Sigma$ satisfying $0 < \lambda_{\min}(\Sigma) \leq \lambda_{\max}(\Sigma) < \infty$, and $N/m \to 0$ as $h \to \infty$ (with $d$ fixed). Standard results for tall random matrices with independent Gaussian columns, see Vershynin (2018), give constants $C, c > 0$ such that, for every $\tau \in (0,1)$, with probability at least $1 - 2\exp(-c\tau^2 m)$,

$$\sqrt{m}\,\sqrt{\lambda_{\min}(\Sigma)}\left(1 - C\sqrt{N/m} - \tau\right) \;\leq\; \sigma_{\min}(\mathbf{A}_{\mathrm{cat}}) \tag{29}$$

$$\leq\; \sigma_{\max}(\mathbf{A}_{\mathrm{cat}}) \tag{30}$$

$$\leq\; \sqrt{m}\,\sqrt{\lambda_{\max}(\Sigma)}\left(1 + C\sqrt{N/m} + \tau\right). \tag{31}$$

Define $\varepsilon_m := C\sqrt{N/m} + \tau$; then $\varepsilon_m \to 0$ as $m = hd \to \infty$ since $N/m \to 0$.

**Step 4: Plug the concentration into eq. (28).**   Using eq. (31) in eq. (28) and writing $m = hd$, we obtain, with the same high probability,

$$\kappa(\mathbf{J}) \leq \sqrt{\frac{C_4^2 C_2^2 + m\,\lambda_{\max}(\Sigma)\,(1+\varepsilon_m)^2}{C_3^2 C_1^2 + m\,\lambda_{\min}(\Sigma)\,(1-\varepsilon_m)^2}}$$

$$= \sqrt{\frac{\frac{C_4^2 C_2^2}{m} + \lambda_{\max}(\Sigma)\,(1+\varepsilon_m)^2}{\frac{C_3^2 C_1^2}{m} + \lambda_{\min}(\Sigma)\,(1-\varepsilon_m)^2}}\;. \tag{32}$$

Letting $h \to \infty$, so that $m \to \infty$, forces $\frac{C_4^2 C_2^2}{m}, \frac{C_3^2 C_1^2}{m} \to 0$ and $\varepsilon_m \to 0$, hence

$$\limsup_{h \to \infty}\; \kappa(\mathbf{J}) \;\leq\; \sqrt{\frac{\lambda_{\max}(\Sigma)}{\lambda_{\min}(\Sigma)}}\;. \tag{33}$$

Since $\kappa(\mathbf{J}) \geq 1$ always, eq. (33) shows that, in the general (non–isotropic) case, the limiting upper bound is $\sqrt{\kappa(\Sigma)}$.

**Step 5: Isotropic specialization $\Rightarrow$ limit 1.**   If, moreover, the columns of $\mathbf{A}_{\mathrm{cat}}$ are *isotropic*—i.e., $\Sigma = \sigma^2 I_N$ for some $\sigma^2 > 0$ (in particular, $\lambda_{\max}(\Sigma) = \lambda_{\min}(\Sigma) = \sigma^2$) then eq. (32) simplifies to

$$\kappa(\mathbf{J}) \;\leq\; \sqrt{\frac{\frac{C_4^2 C_2^2}{m} + \sigma^2(1+\varepsilon_m)^2}{\frac{C_3^2 C_1^2}{m} + \sigma^2(1-\varepsilon_m)^2}} \;\;\xrightarrow[m\to\infty]{}\;\; \frac{1+\varepsilon_m}{1-\varepsilon_m} \;\;\xrightarrow[m\to\infty]{}\;\; 1,$$

again with probability tending to 1. Since $\kappa(\mathbf{J}) \geq 1$, we conclude $\kappa(\mathbf{J}) \to 1$ in probability (indeed, with high probability) as $h \to \infty$ with $d$ fixed.

**Asymptotic rate.** In the isotropic case $\Sigma = \sigma^2 I$, the bound derived there gives (with high probability)

$$\kappa(\mathbf{J}) \leq \sqrt{\frac{\frac{C_4^2 C_2^2}{m} + \sigma^2(1+\varepsilon_m)^2}{\frac{C_3^2 C_1^2}{m} + \sigma^2(1-\varepsilon_m)^2}} = \frac{1+\varepsilon_m}{1-\varepsilon_m}\left(1 + o(1)\right) \qquad (m \to \infty).$$

Using the expansion $\frac{1+\varepsilon_m}{1-\varepsilon_m} = 1 + 2\varepsilon_m + O(\varepsilon_m^2)$ and $\varepsilon_m = O\big(\sqrt{N/m}\big)$, we obtain

$$\kappa(\mathbf{J}) = 1 + O\Big(\sqrt{\tfrac{N}{m}}\Big) = 1 + O\Big(\sqrt{\tfrac{N}{hd}}\Big) \qquad \text{as } h \to \infty \text{ with } d \text{ fixed}.$$

$\square$

**Discussion of assumptions.** The assumptions of theorem 3.1 are natural for multi-head attention at (and near) random initialization, and remain reasonable under standard training practices. For the first assumption, each per–head Jacobian $\mathbf{J}(\mathbf{A}_i)$ factors through linear maps $(X \mapsto XW_{Q_i}, XW_{K_i}, XW_{V_i})$ and the row–wise softmax derivative. With common initializations (Xavier/He or orthogonal), the operator norms of $W_{Q_i}, W_{K_i}, W_{V_i}$ are $O(1)$; modern transformers apply LayerNorm so inputs are scale–controlled; and the softmax Jacobian has a uniformly bounded operator norm (depending only on temperature). Thus $\sigma_{\min/\max}(\mathbf{J}(\mathbf{A}_i))$ admit $h$–independent lower/upper bounds $C_1, C_2$. For second assumption on $\mathbf{P}$, we observe that $\mathbf{P}$ is typically initialized with approximately orthonormal columns, for example via QR/orthogonal initialization or Xavier initialization, implying $\sigma_{\min}(\mathbf{P})$ and $\sigma_{\max}(\mathbf{P})$ are both close to 1, giving constants $C_3, C_4$ independent of $h$. For final assumption, with fixed $d$ and independently initialized heads, the $m = hd$ feature columns of $\mathbf{A}_{\mathrm{cat}}$ are linear/nonlinear combinations of Gaussian ingredients (token features after LayerNorm and independent weight columns), and hence are themselves Gaussian with a common covariance $\Sigma$ that, at initialization, is often close to a scalar multiple of the identity. Independence across columns is exact across heads and across channels within a head (different columns of $W_V$), any residual dependence due to the shared input is mild in high dimension, and standard matrix concentration still captures the growth of $\sigma_{\min}(\mathbf{A}_{\mathrm{cat}})$ as $m$ increases.

### A.1.2 FURTHER THEORETICAL INSIGHTS.

In this section, we give the analogue of theorem 3.1 where we fix the number of heads $h$ and allow the dimension $d$ to vary.

**Theorem A.2.** *Fix the number of heads $h \in \mathbb{N}$ and let $d \in \mathbb{N}$ vary. For each head $i = 1, \ldots, h$, let*

$$\mathbf{A}_i \in \mathbb{R}^{N \times d}, \qquad \mathbf{A}_{\mathrm{cat}} := [\mathbf{A}_1, \ldots, \mathbf{A}_h] \in \mathbb{R}^{N \times (hd)}, \qquad \mathbf{P} \in \mathbb{R}^{(hd) \times D},$$

*and define the multi-head output $\mathbf{MA} := \mathbf{A}_{\mathrm{cat}}\mathbf{P} \in \mathbb{R}^{N \times D}$. Let*

$$y := \mathrm{vec}(\mathbf{MA}) \in \mathbb{R}^{ND}, \qquad \theta := [\theta_1; \ldots; \theta_h; \mathrm{vec}(\mathbf{P})] \in \mathbb{R}^{3hDd + Dhd},$$

*where $\theta_i = [\mathrm{vec}(W_{Q_i}); \mathrm{vec}(W_{K_i}); \mathrm{vec}(W_{V_i})] \in \mathbb{R}^{3Dd}$, and define the (row) per-head Jacobians*

$$\mathbf{J}(\mathbf{A}_i) \in \mathbb{R}^{(3Dd) \times (Nd)}.$$

*Let the Jacobian of the multi-head attention be given by*

$$\mathbf{J} := \mathbf{J}(\mathbf{MA}) = \frac{\partial \theta}{\partial y} \in \mathbb{R}^{(3hDd + Dhd) \times (ND)}.$$

*Assume the following bounds hold uniformly in $d$ (for all sufficiently large $d$):*

(B1) *There exist constants $0 < C_1 \leq C_2 < \infty$ such that $C_1 \leq \sigma_{\min}(\mathbf{J}(\mathbf{A}_i))$ and $\sigma_{\max}(\mathbf{J}(\mathbf{A}_i)) \leq C_2$ for all $i$.*

(B2) *There exist constants $0 < C_3 \leq C_4 < \infty$ such that $C_3 \leq \sigma_{\min}(\mathbf{P})$ and $\sigma_{\max}(\mathbf{P}) \leq C_4$.*

(B3) *Writing $m := hd$, the $m$ columns of $\mathbf{A}_{\mathrm{cat}}$ are independent, mean-zero, Gaussian in $\mathbb{R}^N$ with common covariance $\Sigma$ satisfying $0 < \lambda_{\min}(\Sigma) \leq \lambda_{\max}(\Sigma) < \infty$.*

*Then:*

    (i) **Analogue of theorem A.1.**

$$\sigma_{\min}(\mathbf{J}) \geq \sqrt{C_3^2 C_1^2 + \sigma_{\min}(\mathbf{A}_{\text{cat}})^2}, \qquad \sigma_{\max}(\mathbf{J}) \leq \sqrt{C_4^2 C_2^2 + \sigma_{\max}(\mathbf{A}_{\text{cat}})^2},$$

*hence*

$$\kappa(\mathbf{J}) \leq \sqrt{\frac{C_4^2 C_2^2 + \sigma_{\max}(\mathbf{A}_{\text{cat}})^2}{C_3^2 C_1^2 + \sigma_{\min}(\mathbf{A}_{\text{cat}})^2}}.$$

    (ii) **Asymptotic high-probability bound (analogue of first part of theorem 3.1.** *With probability tending to 1 as $d \to \infty$ (with $h$ fixed),*

$$\limsup_{d \to \infty} \kappa(\mathbf{J}) \leq \sqrt{\frac{\lambda_{\max}(\Sigma)}{\lambda_{\min}(\Sigma)}}.$$

    (iii) **Isotropic specialization (analogue of second part of theorem 3.1).** *If the columns of $\mathbf{A}_{\text{cat}}$ are isotropic, i.e. $\Sigma = \sigma^2 I_N$, then with probability tending to 1,*

$$\kappa(\mathbf{J}) \longrightarrow 1 \qquad \text{as } d \to \infty \text{ with } h \text{ fixed,}$$

*and, moreover,*

$$\kappa(\mathbf{J}) = 1 + O\Big(\sqrt{\tfrac{N}{hd}}\Big).$$

*Proof.* The proof is identical in structure to how we proceeded with theorem A.1 and theorem 3.1, with $m := hd$ now growing via $d \to \infty$ (and fixed $h$).

*Step 1.* An analogue of lemma 3.1 yields the two-block bounds

$$\sigma_{\min}(\mathbf{J}) \geq \sqrt{\sigma_{\min}(\mathbf{P})^2 \big( \min_i \sigma_{\min}(\mathbf{J}(\mathbf{A}_i)) \big)^2 + \sigma_{\min}(\mathbf{A}_{\text{cat}})^2} \tag{34}$$

$$\sigma_{\max}(\mathbf{J}) \leq \sqrt{\sigma_{\max}(\mathbf{P})^2 \big( \max_i \sigma_{\max}(\mathbf{J}(\mathbf{A}_i)) \big)^2 + \sigma_{\max}(\mathbf{A}_{\text{cat}})^2}. \tag{35}$$

*Step 2 (Uniform bounds).* By (B1)–(B2), for all large $d$,

$$C_1 \leq \sigma_{\min}(\mathbf{J}(\mathbf{A}_i)) \leq \sigma_{\max}(\mathbf{J}(\mathbf{A}_i)) \leq C_2, \qquad C_3 \leq \sigma_{\min}(\mathbf{P}) \leq \sigma_{\max}(\mathbf{P}) \leq C_4,$$

which gives the deterministic bound in (i).

*Step 3 (Random-matrix control).* By (B3) and standard Gaussian singular value estimates for tall matrices with $m = hd$ independent columns, with high probability

$$\sqrt{m\,\lambda_{\min}(\Sigma)}\,(1 - C\sqrt{N/m} - \tau) \leq \sigma_{\min}(\mathbf{A}_{\text{cat}}) \leq \sigma_{\max}(\mathbf{A}_{\text{cat}}) \leq \sqrt{m\,\lambda_{\max}(\Sigma)}\,(1 + C\sqrt{N/m} + \tau),$$

for universal constants $C, c > 0$ and any $\tau \in (0, 1)$ (with probability $\geq 1 - 2e^{-c\tau^2 m}$).

*Step 4 (Asymptotics).* Plugging these into the bound from Step 2 and letting $m = hd \to \infty$ yields (ii). In the isotropic case $\Sigma = \sigma^2 I$, the ratio simplifies to $\frac{1 + \varepsilon_m}{1 - \varepsilon_m}$ with $\varepsilon_m = O(\sqrt{N/m})$, giving

$$\kappa(\mathbf{J}) = 1 + O\Big(\sqrt{\tfrac{N}{m}}\Big) = 1 + O\Big(\sqrt{\tfrac{N}{hd}}\Big),$$

which proves (iii). $\qquad\qquad\square$

## A.2 EXPERIMENTS

### A.2.1 VISION TRANSFORMERS ON IMAGENET-1K

**MLP width.** We now consider variations of the hidden-layer size of the MLPs inside a ViT-B model, as an alternative strategy to affect the width of the model. The original model uses a size of $768 \times 4 = 3,072$, where 768 is the token embedding size and 4 is referred to as the "MLP ratio". We train models with a ratio between 1 and 8. fig. 6 shows a limited impact on accuracy that contrasts with the clear large effects of the number of heads from fig. 2. This agrees with the hypothesis made in section 3.3 that MLPs are likely to be already well-conditioned and do not benefit in this regard as much as attention blocks in transformers.

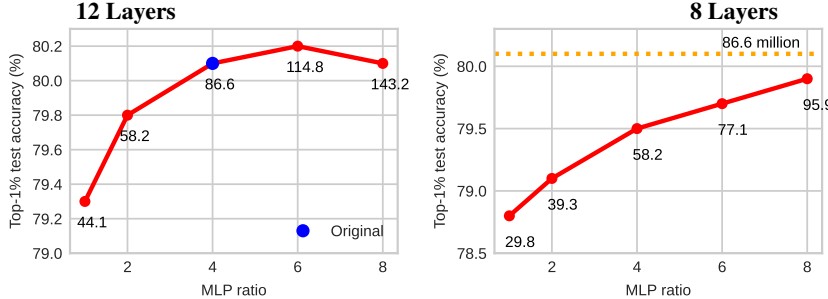

Figure 6: Similar experiments as fig. 2, where each model is now a variant of ViT-B with a **different MLP width** (X axes, reported as a factor of the token-embedding size). According to our predictions, increasing the width of MLPs has a weaker effect than adding attention heads. The slight benefit observed with 12 layers (left) cannot compensate for a reduction of depth to 8 layers (right), unlike what was observed with additional heads in fig. 2.

**Best configurations.** We evaluate additional configurations with depths below 8 in fig. 7. We adjust the number of heads to match the accuracy of the original ViT-B ($\geq 80.1\%$). All configurations still use much fewer parameters than the original model with a better accuracy.

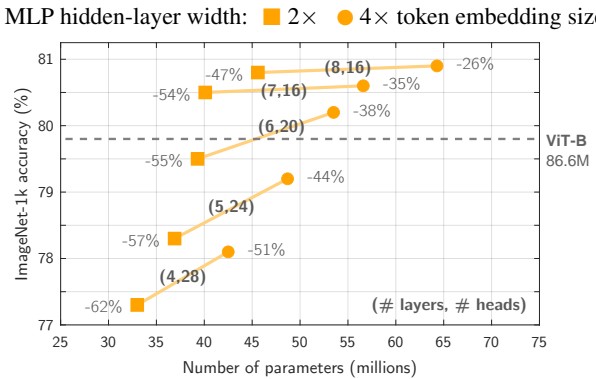

Figure 7: Additional variants of ViT-B with different numbers of layers and heads, and MLP width. Each model is annotated with its reduction in parameters. For 6—8 layers, doubling the MLP width yields little benefit, indicating that the number of heads is more important.

**Detailed results for vision transformers** In section 4.1.2, we demonstrated that several base vision transformers from the literature, ranging from 60 to 90 million parameters, benefit from our approach of increasing the number of heads in each attention layer while reducing the overall depth. In every instance, our configuration performed on par with or better than the original architecture while significantly lowering both parameter count and memory usage (see fig. 4). The detailed configurations are provided in table 4.

We also showed that our methodology could be applied to larger vision transformers with roughly 180-200 million parameters (fig. 5). The configurations for these larger ViTs are given in table 5.

**Hardware and implementation.** All models were trained on 8 Nvidia A100 GPUs using the code base from huggingface: https://github.com/huggingface/pytorch-image-models. Note that we couldn't find an implementation of a TNT large architecture in this code base and that is why we did not have TNT large in our analysis for large vision transformers. The training of each vision transformer architecture we considered follows the original papers cited in section 4.1.

### A.2.2 LONG RANGE SEQUENCE MODELING

We evaluate our approach on Nyströmformers (Xiong et al., 2021b), a transformer-like architecture that uses an approximation of the self-attention with better computational complexity. Our objective is to evaluate the relevance of our findings to an architecture that slightly departs from the original

Table 4: Detailed configurations for a variety of base vision transformers from the literature. Increasing the heads and reducing depth (green) we obtain several transformers that outperform their original counterparts (red) with less parameters and less memory for training.

| (Depth, Heads) | MLP dim. | Top-1% Acc. | Top-5% Acc. | Params. (millions) | Memory (GB) |
|---|---|---|---|---|---|
| **ViT-B on ImageNet-1k** | | | | | |
| (12, 12) | 3072 | 80.1 | 94.2 | 86.6 | 178.4 |
| (7,16) | 1536 | 80.4 | 94.9 | 40.1 | 101.6 |
| **DeiT-B on ImageNet-1k** | | | | | |
| (12, 12) | 3072 | 80.4 | 95.1 | 86.6 | 178.4 |
| (7,16) | 1536 | 80.8 | 95.3 | 40.1 | 101.6 ↓ |
| **XCiT-Medium on ImageNet-1k** | | | | | |
| (24, 8) | 2048 | 81.4 | 95.5 | 84.4 | 320.8 |
| (12,16) | 2048 | 81.7 | 95.6 | 59.0 | 196 |
| **TNT-B on ImageNet-1k** | | | | | |
| (12, 10) | 2560 | 82.3 | 95.7 | 65.4 | 266.4 |
| (8,16) | 2560 | 82.3 | 95.8 | 30.9 | 200.8 |
| **VOLO-d3 on ImageNet-1k** | | | | | |
| ([8, 8, 16, 4], [8, 16, 16, 16]) | (1024, 2048, 2048, 2048) | 82.6 | 95.6 | 86 | 209.2 |
| ([4, 4, 8, 2], [16, 32, 32, 32]) | (768, 1536, 1536, 1536) | 82.6 | 95.7 | 47.5 | 144.8 |
| **DaViT-B on ImageNet-1k** | | | | | |
| ([1,1,9,1], [4, 8, 16, 32]) | (512, 1024, 2048, 4096) | 83.3 | 96.0 | 88.0 | 294.4 |
| ([1, 1, 5, 1], [4, 8, 32, 32]) | (512, 1024, 2048, 4096) | 83.5 | 96.1 | 62.0 | 233.6 |

Table 5: Detailed configurations for a variety of large vision transformers from the literature. Increasing the heads and reducing depth (green) we obtain several transformers that outperform their original counterparts (red) with less parameters and less memory for training.

| (Depth, Heads) | MLP dim. | Top-1% Acc. | Top-5% Acc. | Params. (millions) | Memory (GB) |
|---|---|---|---|---|---|
| **ViT-L on ImageNet-1k** | | | | | |
| (24, 16) | 4096 | 80.6 | 94.4 | 203.6 | 200.0 |
| (8,30) | 2048 | 81.1 | 95.1 | 98.6 | 115.2 |
| **DeiT-L on ImageNet-1k** | | | | | |
| (24, 16) | 4096 | 81.5 | 95.3 | 203.6 | 200.0 |
| (8,30) | 2048 | 81.8 | 95.4 | 98.6 | 115.2 ↓ |
| **XCiT-L on ImageNet-1k** | | | | | |
| (24, 16) | 3072 | 82.1 | 95.9 | 189.1 | 275.2 |
| (12,24) | 3072 | 82.4 | 95.9 | 103.8 | 160.8 |
| **VOLO-d4 on ImageNet-1k** | | | | | |
| ([8, 8, 16, 4], [12, 16, 16, 16]) | (1536, 3072, 3072, 3072) | 83.0 | 96.1 | 193.0 | 294.4 |
| ([4, 4, 8, 2], [24, 32, 32, 32]) | (768, 1536, 1536, 1536) | 83.1 | 96.2 | 105.6 | 188.8 |
| **DaViT-L on ImageNet-1k** | | | | | |
| ([1,1,9,1], [6, 12, 24, 48]) | (768, 1536, 3072, 6144) | 83.6 | 96.5 | 196.8 | 238.4 |
| ([1, 1, 5, 1], [6, 12, 48, 48]) | (768, 1536, 3072, 6144) | 83.6 | 96.6 | 140.0 | 186.4 |

transformer architecture of Vaswani (2017). Nyströmformers are well suited to long sequences and we therefore evaluate them on the Long-Range Arena (LRA) benchmark (Tay et al., 2021).

Our base model follows the original paper Xiong et al. (2021b) and uses 2 layers and 2 attention heads per layer. We also train variants with 2-8 heads and 1-2 layers. The results on the ListOps task (see fig. 8) and the Text classification task (see fig. 9) show that additional heads increase the accuracy. This allows reducing the depth to a single layer while improving its accuracy. These results hold across other tasks of the LRA benchmark (see table 6).

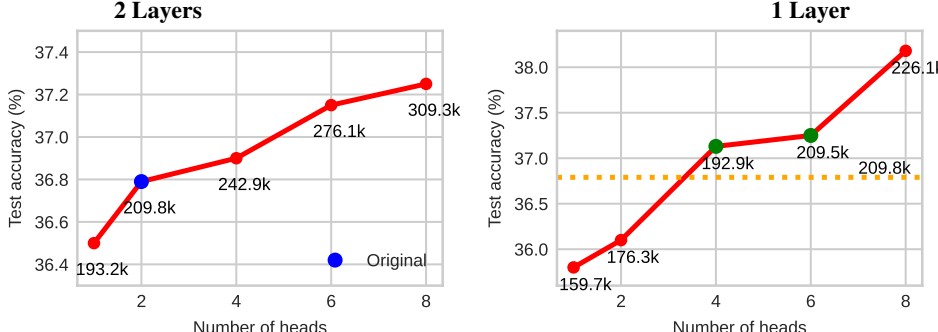

Figure 8: Accuracy on the ListOps task of the LRA benchmark with variants of the Nyströmformer. The original model from Xiong et al. (2021b) uses 2 layers (left) and we also evaluate models with a single layers (right). Each model is annotated with its total number of parameters. According to our predictions, the number of heads correlates with performance. Remarkably, our models with just 1 layer and $\geq 4$ heads (green dots) all obtain a **higher test accuracy with fewer parameters** than the original model (dotted line).

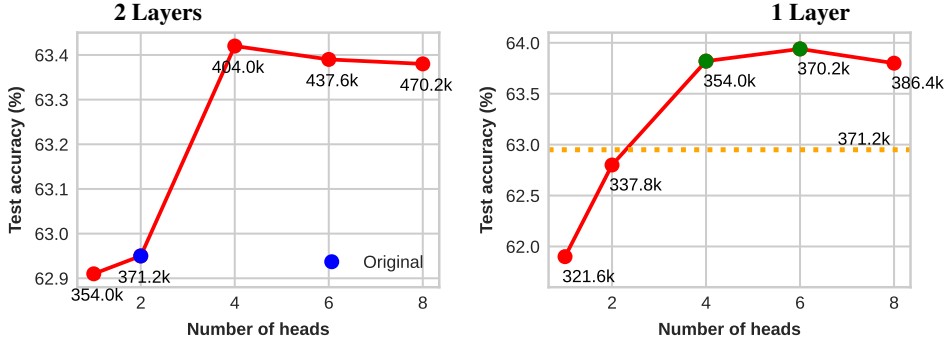

Figure 9: Accuracy on the text classification task of the LRA benchmark with variants of the Nyströmformer. The original model from Xiong et al. (2021b) uses 2 layers (left) and we also evaluate models with a single layers (right). Each model is annotated with its total number of parameters. According to our predictions, the number of heads correlates with performance. Remarkably, our models with just 1 layer and $\geq 4$ heads (green dots) all obtain a **higher test accuracy with fewer parameters** than the original model (dotted line).

**Hardware and implementation.** The Nyströmformer experiments carried out on one Nvidia A6000 GPU. The implementation followed the original paper of Xiong et al. (2021b) and its GitHub repo (Xiong et al., 2021a).

Table 6: Evaluation of variants of the Nyströmformer (Xiong et al., 2021b) on different datasets of the Long-Range Arena (LRA) benchmark (Tay et al., 2021). We compare the original model (2 layers, 2 heads) with our variant (1 layer, 4 heads). On every task, it outperforms the original model with the same number or slightly fewer parameters.

| ListOps | | | |
|---|---|---|---|
| (Depth, heads) | Top-1% Acc. | Parameters | |
| (2, 2) | 36.79 | 209.8k | |
| (1,4) | **37.13** | 192.9k | (-9%) |
| **Text Classification** | | | |
| (Depth, heads) | Top-1% Acc. | Parameters | |
| (2, 2) | 62.95 | 371.2k | |
| (1,4) | **63.82** | 354.0k | (-5%) |
| **Document Retrieval** | | | |
| (Depth, heads) | Top-1% Acc. | Parameters | |
| (2, 2) | 79.3 | 394.8k | |
| (1,4) | **79.5** | 394.8k | (same) |
| **Image Classification** | | | |
| (Depth, heads) | Top-1% Acc. | Parameters | |
| (2, 2) | 37.2 | 191.2k | |
| (1,4) | **38.2** | 191.2k | (same) |
| **Pathfinder** | | | |
| (Depth, heads) | Top-1% Acc. | Parameters | |
| (2, 2) | 69.8 | 190.2k | |
| (1,4) | **69.9** | 190.2k | (same) |

