# OpenReview forum: "Leaner Transformers: More Heads, Less Depth"
_ICLR.cc/2026/Conference — ICLR 2026 Conference Withdrawn Submission_

### Official Review · Reviewer_QqCC · 2025-10-28

**Soundness:** 2
**Presentation:** 2
**Contribution:** 2
**Rating:** 2
**Confidence:** 3

**Summary:**

The paper primarily investigates the role of attention heads in Transformer architectures, analyzing how the number of heads affects model performance across different Transformer variants. It explores the trade-off between the number of heads and the number of layers, demonstrating that appropriately scaling these components can lead to improved performance and higher accuracy. Furthermore, the paper shows that increasing the number of heads enhances the conditioning of the Jacobian of the attention matrix, an effect supported by empirical results on Vision Transformers (ViT). The authors also conduct experiments on different domains like Vision Transformer architectures as well as GPT and LRA benchmarks.

**Strengths:**

### Strengths

The paper have multiple strengths:

* **Clarity in Presentation:** The main results are presented clearly and effectively, with key findings highlighted in well-designed boxes that make them easy to follow and visually accessible.
* **Jacobian Analysis:** The analysis of the Jacobian matrix of the attention mechanism is a valuable and insightful direction. It provides a deeper understanding of the conditioning of the attention matrix and aligns well with the objectives outlined in the related literature.
* **Comprehensive Vision Experiments:** The paper conducts extensive and thorough experiments on Vision Transformer models, including both base and large-scale variants, demonstrating a strong experimental design and analysis.

**Weaknesses:**

While the experimental setup in the paper is extensive, there are several notable shortcomings:

**1) Simplistic Preliminary Section**
The preliminary section is overly simplified. For instance, *Equation (2)* omits the layer normalization steps between layers. Additionally, in the paragraph following *Equation (3)*, the similarity metric for attention is denoted as $\phi$, which should ideally be expressed as $\phi(q, k)$. However, the paper describes it as *softmax($\phi(q, k)$)*, a formulation that lacks clarity and may lead to ambiguity in interpretation.

---

**2) “Why Scale the Number of Depths?”**
This section argues that increasing model depth leads to richer feature representations. Although this is intuitively reasonable, the authors provide neither visualizations nor quantitative evidence to substantiate the claim. For example, visualizations of attention maps could demonstrate whether additional heads actually learn more diverse or complementary features. Furthermore, while deeper models can indeed capture higher-level abstractions, the paper does not rigorously analyze or quantify this effect.

---

**3) Increasing the Number of Heads leads to more parameters**
As mentioned on page 5, increasing the number of attention heads substantially raises the model’s parameter count. To better understand the trade-off between *depth* and *heads*, the authors could conduct experiments where model depth is increased while keeping the number of heads constant, so maintaining approximately the same parameter budget. Such an ablation would provide clearer insights into whether performance gains arise from additional heads or simply from a larger model capacity.

---

**4) Weakness of the LRA Benchmark**
The Long Range Arena (LRA) benchmark is no longer considered a strong evaluation framework for transformers. Recent works have demonstrated that State Space Models (SSMs) achieve significantly higher accuracy on these tasks, while transformers can perform well when pre-trained with self-supervised objectives (see [1]). Moreover, improvements reported on LRA tasks are often marginal and may not reflect meaningful advancements in model architecture.

---

**5) Incomplete Experimental Comparisons**
Although the experiments span multiple tasks, they lack comprehensive one-to-one comparisons. The study focuses primarily on increasing the number of heads, without adequately comparing this to scaling model *depth* or *width*. A more systematic analysis—both numerical and theoretical, would strengthen the conclusions and provide a deeper understanding of how these factors influence model performance.

---

**Typos and Formatting Issues**

* *Equation (11)* contains a typographical error in the parentheses, one bracket is larger than the other.
* In *Section 4.1.1*, the patch size is written as *16!x!16*, which appears to be a formatting error (the exclamation marks should be removed).

---

### Reference

[1] Never Train from Scratch: Fair Comparison of Long-Sequence Models Requires Data-Driven Priors  Ido Amos, Jonathan Berant, Ankit Gupta

**Questions:**

The main questions are mentioned in the weakness section but can be summarized as follows:

1) Is there any theoretical proof that, under the same parameter regime, increasing depth or the number of heads is more beneficial than the other?

2) If not, are there any experimental results showing that increasing the number of heads (which leads to more parameters) performs better than increasing the depth? A one-to-one comparison would help clarify how scaling depth versus the number of heads contributes to the model’s overall performance.

---

### Official Review · Reviewer_5iHU · 2025-10-31

**Soundness:** 3
**Presentation:** 3
**Contribution:** 3
**Rating:** 4
**Confidence:** 3

**Summary:**

This paper revisits the overparameterization trend in Transformers and proposes a new design principle: “More heads, less depth.” The authors provide a theoretical analysis showing that increasing the number of attention heads improves the conditioning of the attention Jacobian, which should stabilize optimization. They demonstrate empirically that one can reduce network depth while maintaining or improving accuracy, achieving 30–50% parameter reduction across vision (ImageNet), language (GLUE, TinyStories), and long-context (LRA) benchmarks.

**Strengths:**

- Provides a mathematically grounded reinterpretation of multi-head attention’s role in optimization.
- Consistent empirical validation across domains (vision, NLP, long-sequence).
- Strong practical message toward leaner Transformer design.
- The combination of theoretical insight and empirical validation across different domains makes the findings broadly relevant and practically actionable.

**Weaknesses:**

- Core assumptions (independent Gaussian heads, isotropy, bounded singular values) are unrealistic during actual training; their validity beyond initialization is unclear.
- The theoretical link between Jacobian conditioning and global optimization/generalization remains unproven.
- Experiments are limited to mid-scale models (<200M parameters); it remains unclear whether the same depth-head trade-off persists under large-scale pretraining or diverse data distributions.

**Questions:**

- Can the authors empirically measure Jacobian conditioning or NTK spectra after training to verify that it remains improved, not just at initialization?
- Can you provide quantitative results linking improved Jacobian conditioning to faster convergence or better generalization performance?
- As model and data scales grow, the assumptions of independent Gaussian heads, isotropy, and bounded Jacobian singular values may become less valid. How do the authors expect these assumptions and the resulting conditioning behavior to evolve in larger Transformer regimes?

---

### Official Review · Reviewer_NMme · 2025-10-31

**Soundness:** 4
**Presentation:** 3
**Contribution:** 4
**Rating:** 8
**Confidence:** 4

**Summary:**

The authors first present theoretical results that show that increasing the number of heads in a transformer improves the conditioning of the Jacobian of the attention block. They then design new architectures with more heads and fewer layers to exploit this insight, reducing parameter count whilst still maintaining high accuracy across a range of tasks that include vision, language, and sequence modelling.

**Strengths:**

1. The mathematical and empirical analysis is clearly presented
2. The mathematical analysis generates interesting insights which then strongly motivate the architectural changes explored
3. The empirical analysis is extensive, covering several different domains and task types
4. The empirical analysis is detailed, aiding reproducibility, and conducts lots of different experiments to validate the hypotheses of the paper
5. The empirical results are strong, and give good evidence to the hypotheses of the paper

**Weaknesses:**

1. The authors may wish to cite further prior empirical work showing that one can make this tradeoff between attention head count and layer size while still preserving accuracy, such as https://arxiv.org/abs/2210.00640, or https://proceedings.neurips.cc/paper_files/paper/2023/hash/3504a4fa45685d668ce92797fbbf1895-Abstract-Conference.html
2. Given the dominance of scaling GPT-style transformers and language-modelling in today's AI landscape, it would be interesting to see how the improvements from this architectural change are affected when used in larger LMs. A plot or analysis of train loss on a large language corpus across model sizes from 100s of millions to 10s of billions parameters would be very interesting to see.

#### Typos (Score given assuming these will all be fixed):
1. It's not material to the rest of the paper, but equation (1) should be (ignoring normalisation): $\mathbf{T}(X) = \mathbf{F}(\mathbf{A}(X) + X) + \mathbf{A}(X) + X$
2. On line 146/147 it's stated "and $h$ and thus $h$ must divide $D$." Presumable one of these $h$'s should be $d$?
3. Line 291, "$16! \times !16$", presumably the !s shouldn't be there?
4. Line 338/339 "number of heads (left) ... head dimension (left)". Both plots list number of heads on the x-axis. Same patter for figure 3 (line 357). These captions and plot labels are inconsistent with the main text, though it can be easily inferred what is meant (hence typo classification for these errors)
5. Line 367/368 "Row is the standard ViT-B..." which row?

**Questions:**

1. For the language modelling experiments, how do you think your results will be affected as the model size is scaled up into billions of parameters?

---

### Official Review · Reviewer_qDS2 · 2025-11-01

**Soundness:** 2
**Presentation:** 2
**Contribution:** 2
**Rating:** 2
**Confidence:** 4

**Summary:**

This paper challenges the convention that larger vision transformers are required for strong performance and shows that many transformers are unnecessarily oversized. The authors present a theoretical principle that redefines the role of multi-head attention and redesign popular architectures with more heads and fewer layers. Empirical results shows that this trade-off reduces parameter counts by up to 30--50% while preserving accuracy, and experiments on multiple tasks demonstrate the effectiveness of the proposed method.

**Strengths:**

1. The direction of reducing the depth of vision transformers seems interesting and indeed worths exploration. The depth of deep networks plays a more essential role compared to width in terms of practical efficiency and it is appreciated that the authors have provided theoretical insights to validate the proposed scheme.
2. The writing is clear and the paper is easy to follow. The overall structure is well organized and the idea is presented in a coherent manner.
3. It is appreciated that the authors have conduct experiments on different backbones and various tasks, spanning from computer vision to language and sequence modeling.

**Weaknesses:**

1. The technical contribution of the paper seems limited. The core idea, i.e., more heads and fewer layers leads to better trade-off between efficiency and accuracy, looks like the empirical finding which can hardly be regarded a systematic methodology. Although the authors have provided theoretical insights, it does not provide a systematic guideline of how to adjust the trade-off between heads and layers. Is the trade-off chosen from empirical results per architecture?
2. While the rationale has been validated on high level understanding tasks, its efficacy on dense prediction tasks like sementic segmentation or object detection remains unverified. It would better if the authors could validate the proposed method on dense prediction tasks as well.
3. The ablation is missing. It seems that the authors have different criterion on choosing the trade-off between heads and layers for different specific models, and the ablation on this part is missing.
4. Although the authors have shown advantage of the proposed method in model parameters and memory usage, the practical inference speedup is unknown and it is better to compare the CPU and GPU latency of the proposed architecture with the conventional structure.

**Questions:**

see weakness above

---

### Official Review · Reviewer_pNzE · 2025-11-01

**Soundness:** 3
**Presentation:** 3
**Contribution:** 3
**Rating:** 6
**Confidence:** 4

**Summary:**

This paper takes a step back and questions design habit of Transformers: the tendency to scale mainly by depth.
The authors develop a simple argument that increasing the number of attention heads improves the conditioning of the attention Jacobian, which in turn stabilize optimization. Based on that, they explore a trade-off between heads and layers. They show that, for the same FLOPs, a shallower network with more heads can match or even slightly exceed the performance of deeper ones. They demonstrate this on ImageNet (ViT), GLUE, TinyStories, and LRA benchmarks.

**Strengths:**

- The theoretical idea is fresh and intuitive. It offers a new perspective on why multi-head setups are easier to train than single-head variants.
- The empirical validation is broad. I like that the authors show the results in both vision and language tasks. The Imagenet and GLUE results are convincing. The savings in memory and parameters are impressive.
- Many transformer related papers are technically heavy and having unclear design guidance. But this paper contributes a simple rule of thumb that practitioners can test immediately.

**Weaknesses:**

- The theoretical section assumes independence and isotropy among heads, which probably doesn't hold in trained models. It would help to at least show empirical correlation plots or conditioning statistics from real networks to bak this up.
- The experiments are all in the mid-scale regime (ViT-B/L, BERT-base size). I would have liked to see a preliminary results on a > 1B-parameter model to check whether the rule survives modern LLM training dynamics.
- I'm not entirely convinced the performance improvements stem directly from the better conditioning. It could just be that redistributing parameters across heads changes the inductive bias or regularization behavior. Some ablations around learning rate or normalization might clarify that.
- The paper could discuss inference speed a bit more carefully. Fewer layers should reduce sequential latency, but many-head attention can be bandwidth-limited.

**Questions:**

1. You assume independence and isotropy among head outputs in the conditioning analysis. Could you include empirical evidence, e.g., inter-head correlation or Jacobian spectra from trained models, to verify that real Transformers approximately satisfy these assumptions?
2. Have you tried this head–depth trade-off on larger models (hundreds of millions to billions of parameters)? It would be interesting to see whether the same effect holds once training dynamics and regularization at scale come into play.
3. Could the observed benefit come from changed regularization or parameter distribution rather than better Jacobian conditioning? For instance, have you compared against simply redistributing parameters into wider MLP blocks?
4. Prior work (e.g., Michel et al., 2019) showed that many heads can be removed post-training with little effect. How do you reconcile your finding that additional heads help optimization with evidence that many heads are redundant at convergence?

Reference:
[Michel et al., 2019] Analyzing Multi-Head Self-Attention: Specialized Heads Do the Heavy Lifting, the Rest Can Be Pruned.

---

### Note · Authors · 2025-11-25

I have read and agree with the venue's withdrawal policy on behalf of myself and my co-authors.